# Dual Inhibition of HIV-1 and Cathepsin L Proteases by *Sarcandra glabra*

**DOI:** 10.3390/molecules27175552

**Published:** 2022-08-29

**Authors:** Bowen Pan, Sumei Li, Junwei Xiao, Xin Yang, Shouxia Xie, Ying Zhou, Jian Yang, Ying Wei

**Affiliations:** 1College of Pharmacy, Guizhou University of Traditional Chinese Medicine, Guiyang 550025, China; 2Department of Pharmacology, Shenzhen People’s Hospital (The Second Clinical Medical College, Jinan University, The First Affiliated Hospital, Southern University of Science and Technology), Shenzhen 518020, China; 3College of Pharmacy and Nutrition, University of Saskatchewan, 107 Wiggins Road, Saskatoon, SK S7N 5E5, Canada

**Keywords:** *Sarcandra glabra*, HIV-1 protease, Cathepsin L protease, inhibition, UPLC-HRMS, molecular docking

## Abstract

The COVID-19 pandemic continues to impose a huge threat on human health due to rapid viral mutations. Thus, it is imperative to develop more potent antivirals with both prophylactic and treatment functions. In this study, we screened for potential antiviral compounds from *Sa**rcandra glabra* (SG) against Cathepsin L and HIV-1 proteases. A FRET assay was applied to investigate the inhibitory effects and UPLC-HRMS was employed to identify and quantify the bioactive components. Furthermore, molecular docking was carried out to get a glimpse of the binding of active compounds to the proteases. Our results showed that the **SG** extracts (SGW, SG30, SG60, and SG85) inhibited HIV-1 protease with an IC_50_ of 0.003~0.07 mg/mL and Cathepsin L protease with an IC_50_ of 0.11~0.26 mg/mL. Fourteen compounds were identified along with eight quantified from the **SG** extracts. Chlorogenic acid, which presented in high content in the extracts (12.7~15.76 µg/mg), possessed the most potent inhibitory activity against HIV-1 protease (IC_50_ = 0.026 mg/mL) and Cathepsin L protease (inhibition: 40.8% at 0.01 mg/mL). Thus, **SG** extracts and the active ingredients could potentially be used to prevent/treat viral infections, including SARS-CoV-2, due to their dual-inhibition functions against viral proteases.

## 1. Introduction

It has been more than two and a half years since the start of the COVID-19 pandemic, with more than 563 million people being infected and 6.37 million people being killed globally (accessed on 20 July 2022) [1]. Besides extensive vaccination, the disease is mainly treated with antiviral and immunotherapeutic agents [2,3]. The pandemic has prompted extensive research into finding more potent antiviral therapeutics. Our laboratory has been working on identifying and developing antiviral agents from traditional medicinal plants. Recently, we showed that extracts from the insect gall of *Hypericum kouytchense* inhibited both Cathepsin L and HIV-1 proteases, and several active ingredients were identified [4]. Cathepsin L protease (Cat L PR) regulates SARS-CoV-2 viral entry by activating the spike protein [5]. This study has inspired us to examine whether other medicinal plants possess dual- and even pan-inhibitory functions against viruses.

*Sarcandra glabra* (Thunb.) Nakai (**SG**, “Jiu Jie Cha”in Chinese) has been used as an “antipyretic-detoxicate drugs” in traditional Chinese Medicine since the Qing Dynasty (AD) [6]. In the southeast region of the Guizhou province, China, ethnic people usually mix *S. glabra* with glutinous rice, peanuts, or soybeans as tea in daily life, as it is capable of reducing fever [7]. The State Food and Drug Administration of China has approved six **SG** water extract based single-drug prescriptions (Zhongjiefeng Injection, Zhongjiefeng Granules, Zhongjiefeng Tablets, XueKang Oral Liquid, XueKang Granules and Qingren Xiaoyanning Capsules) to treat upper respiratory tract infections, pneumonia, gastritis, viral myocarditis, tumor, and thrombocytopenia [8]. It was reported that **SG** extracts could reduce the incidence rate and mortality of restraint stress mice caused by the H_1_N_1_ influenza virus [9]. More than two hundred compounds have been isolated and identified from **SG** extracts, including flavonoid type (astilbin), coumarin type (isofraxidin), phenolic acid type (rosmarinic acid and chlorogenic acid), and sesquiterpenes type (chloranthalactone) molecules [7]. In recent years, isofraxidin and rosmarinic acid-4-*O*-*β*-*d*-glucoside were found to induce anti-influenza effects [9,10]. In addition, sesquiterpenoids from **SG** have shown potent anti-inflammatory functions [11,12]. Taking into consideration that SARS-CoV-2 infection may lead to severe pneumonia and other inflammatory conditions, **SG** might be beneficial in relieving symptoms and even treating the infection in COVID-19 patients. Furthermore, lindenance disesquiterpenoids from *Chloranthus japonicas* exhibited potent anti-HIV-1 activities [12]. Thus, we decided to examine whether SG extracts and their ingredients possess dual-inhibitory functions against Cathepsin L and HIV-1 proteases. 

## 2. Results and Discussion

### 2.1. Inhibition of HIV-1, CatL and Renin PRs by SG Extracts

The inhibitory activity of **SG** extracts towards renin protease was measured in order to evaluate their potential toxicity towards humans. Renin was treated with the **SG** extracts (concentration: 10.0–1000 µg/mL) for 30 min at 37 °C. In general, **SG** extracts displayed medium to high IC_50_ values (ranging from 120 to 1000 µg/mL) against renin, implicating that they are less likely to be toxic to humans. Then, we measured the inhibitory activities of the **SG** extracts towards HIV-1 and CatL PRs. As shown in Table 1, the **SG** extracts (SGW, SG30, SG60, and SG85) showed strong inhibition of HIV-1 PR with IC_50_ values ranging from 0.003 to 0.07 mg/mL. For CatL PR, the IC_50_ values ranged from 0.15 to 0.26 mg/mL. This indicates that the **SG** extracts, especially SGW and SG30, are much more potent towards HIV-1 PR (Appendix A).

### 2.2. Characterization of the Major Ingredients in SG Extracts by UPLC-MS

*Sarcandra glabra* contains many active ingredients, including flavonoids, triterpenoids, organic acids, coumarins, and monoterpene-conjugated dihydrochalcone derivatives [13,14]. Fourteen selected active compounds were profiled in the **SG** extracts using UPLC-MS (Figure 1 and Figure 2, and Table 2). The compounds were betaine, D-(-)-quinic acid, paeonol, protocatechuic acid, chlorogenic acid, 7-hydroxycoumarine, caffeic acid, fraxetin, astilbin, isofraxidin, 3,4-dicaffeoylquinic acid, rosmarinic acid, alpinetin, and artemisinin.

### 2.3. Inhibition of CatL and HIV-1 PRs by the Major Ingredients in SG Extracts

In order to identify the ingredients providing the potent inhibitory activities against the viral proteases, a fluorescence resonance energy transfer assay was carried out with CatL and HIV-1 PRs (Table 3 and Table 4). At a concentration of 0.01 mg/mL, protocatechuic acid, chlorogenic acid, caffeic acid, isofraxidin, 7-hydroxycoumarine, rosmarinic acid, fraxetin, and alpinetin inhibited CatL PR by 11.9~42.1% (Table 3), with protocatechuic acid and chlorogenic acid as the most potent compounds. For HIV-1 PR, seven compounds (chlorogenic acid, 3,4-dicaffeoylquinic acid, caffeic acid, rosmarinic acid, astilbin, alpinetin, and artemisinin) showed more than 50% inhibition at a concentration of 1.0 mg/mL (Table 4). Chlorogenic acid was the most potent compound, exhibiting 100% inhibition of HIV-1 PR. Its IC_50_ value was determined to be 0.026 mg/mL. These results suggest that four ingredients from **SG** extracts, chlorogenic acid, caffeic acid, rosmarinic acid, and alpinetin, could be promising lead compounds in developing dual inhibitors of HIV-1 and CatL PRs (Appendix A).

### 2.4. Molecular Docking Results

To get a glimpse of how the active ingredients in the **SG** extracts inhibit CatL and HIV-1 PRs, we docked them to the active sites of CatL PR (PDB ID: 3OF9) and HIV-1 PR (PDB ID: 1QBS), respectively, using Schrödinger Suite 2021-1. For CatL PR, chlorogenic acid and rosmarinic acid showed reasonably good binding affinity with a docking score less than −6.0 (Table 5). For chlorogenic acid, there were four hydrogen bonds formed: between a hydroxyl group of the caffeic acid moiety and the residues GLY165 and ASP163; between the carboxyl group of the quinic acid moiety and the residue LYS118; and between a hydroxyl group of the quinic acid moiety and the residue ASH72. Additionally, a salt bridge was formed between the deprotonated carboxyl group in the quinic acid moiety and the protonated amidogen in residue LYS118 (Figure 3a). For rosmarinic acid, three hydrogen bonds were formed between hydroxyl and carbonyl groups in the caffeic acid moiety and residues ASP163, GLN20, and CYS26, and two hydrogen bonds were formed between carboxyl and hydroxyl groups in the dihydroxyphenyl lactic acid moiety and residues ASP163 and MET162 (Figure 3b). Although the docking score for the most potent protocatechuic acid was also less than −6.0, its glide emodel was the highest (−26.518 kcal/mol) among the docked molecules. This implies that protocatechuic acid was present in an unfavorable “pose” in the docking force field and further studies, including protein crystallography, may be needed to fully illustrate its mechanism of inhibition.

Towards HIV-1 PR, four compounds showed good binding affinity with docking scores less than −9.0 (Table 6), with 3,4-dicaffeoylquinic acid, rosmarinic acid, and chlorogenic acid as the top three. For 3,4-dicaffeoylquinic acid, three hydroxyl and a carbonyl groups in the dicaffeic acid moieties formed five hydrogen bonds with residues ASP30(A), ASP30(B), GLY48(B), ILE50(B), and ILE50(A), and the deprotonated carboxyl group in the quinic acid moiety formed one salt bridge with the protonated amidogen of residue ARG8(B) (Figure 4a). For rosmarinic acid, two hydroxyl groups and one carbonyl group from the caffeic acid moiety formed five hydrogen bonds with residues ASH25(B), GLY27(B), ASP29(B), ILE50(A), and ILE50(B). The hydroxyl and carbonyl groups of the 3,4-dihydroxylphenylpropionic acid moiety formed two hydrogen bonds with the residues GLY48(A) and ASP25(B) (Figure 4b). Regarding chlorogenic acid, three hydrogen bonds were formed between the caffeic acid moiety and the residues ILE50(A), ILE50(B), and ASP30(B), and three hydrogen bonds were formed between the quinic acid moiety and the residues ASP25(A) and ASH25(B) (Figure 4c). In general, the docking results support the enzyme inhibition studies, with chlorogenic acid as a potent inhibitor of CatL PR, and 3,4-dicaffeoylquinic acid and chlorogenic acid as potent inhibitors of HIV-1 PR.

### 2.5. Quantitation of Antiviral Ingredients in SG Extracts by UPLC-MS

To evaluate potential dose-effect relationships, eight compounds showing inhibitory effects against Cat L and/or HIV-1 PRs (**4**–**5**, **7**–**8**, **10**–**13**) were quantified in the different **SG** extracts. As shown in Table 7, the contents of protocatechuic acid, chlorogenic acid (Appendix A), 3,4-dicaffeoylquinic acid, and rosmarinic acid were higher than 5.0 µg/mg in SGW. Rosmarinic acid was the most abundant ingredient in the **SG** extracts at 21.242 μg/mg in SGW, 22.804 μg/mg in SG30, 27.730 μg/mg in 60% SG60, and 29.943 μg/mg in 85% SG85. Furthermore, protocatechuic acid was predominantly present in SGW (14.94 μg/mg).

In the present study, we observed moderate to strong inhibitory activities of the **SG** extracts (SGW, SG30, SG60, and SG85) towards CatL and HIV-1 PRs, with very low toxicity against human renin protease. In total, 14 compounds were identified and confirmed from **SG** extracts, among which chlorogenic acid and rosmarinic acid were the major compounds showing dual inhibitions of HIV-1 and CatL PRs. For CatL PR, the extracts showed moderate inhibitions with IC_50_ values ranging between 0.15 and 0.26 mg/mL. At a concentration of 0.01 mg/mL, three coumarins, fraxetin, isofraxetin, and 7-hydroxycoumarine, showed moderate inhibition (15.0~27.0%). One flavanone alpinetin showed a stronger inhibition than flavanonol glycoside astilbin of CatL PR (11.9% vs. −24.7%). A comparison of 3,4-dihydroxycinnamic acid derivatives (**5**, **7**, **11,** and **12**), quinolinic acid (**2**) without a caffeic acid moiety in its structure, and 3,4-dicaffeoylquinic acid (**11**) with di-3,4-dihydroxycinnamic acid groups in its structure showed no inhibition of CatL PR, suggesting that the mono 3,4-dihydroxycinnamic acid group deserves further investigation in developing more potent inhibitors. For HIV-1 PR, chlorogenic acid, 3,4-dicaffeoylquinic acid, caffeic acid, and rosmarinic acid exhibited strong inhibitory effects with IC_50_ values of 0.026, 0.13, 0.48, and 0.54 mg/mL, respectively. Potent anti-HIV-1 PR activity of the above compounds was further supported by molecular docking. As outlined in Table 1 and Table 7, SGW (IC_50_ 0.003 mg/mL) and SG30 (IC_50_ 0.008 mg/mL) exhibited similar inhibitory potency towards HIV-1 PR, but about 10–20 folds more potency than SG60 and SG85 (respective IC_50_ values of 0.063 and 0.07 mg/mL). This is highly likely due to the higher contents of chlorogenic acid, rosmarinic acid, and 3,4-dicaffeoylquinic acid in SGW and SG30 extracts, as well as protocatechuic acid being predominantly present in SGW (14.94 μg/mg). Previous pharmacological studies also show that protocatechuic acid, chlorogenic acid, caffeic acid, and rosmarinic acid possess anti-inflammatory activity and that their derivatives can significantly reduce paraquat-induced activation of inflammatory factors in lung tissue [15,16]. Our current study suggests that **S****G** extracts and their active ingredients may possess dual- and pan-inhibitory functions against viral proteases. 

## 3. Materials and Methods

### 3.1. Plant Material and Extraction

**SG** was purchased at Guiyang Wan Dong Medicine Market (Guiyang, Guizhou, China) in October 2020 and identified by Professor De-yuan Chen, a botanist at Guizhou University of Traditional Chinese Medicine (GUTCM). A specimen of the plant (voucher no. 202010/SG) is deposited at GUTCM. Four dried and finely powdered samples of **SG** (200 g) were refluxed twice (1:6, *v*/*v*) with H_2_O, 30% MeOH, 60% MeOH, and 85% MeOH, respectively, for 1 h. For each type of extraction, the resulting extracts were filtered, concentrated under reduced pressure, and vacuum dried. 

### 3.2. Reagents and Instruments

LC-MS grade reagents acetonitrile and formic acid, and HPLC grade reagents isopropanol and methanol were purchased from Rhawn Reagent Co. Ltd. (Shanghai, China). Dimethylsulfoxide (DMSO) was purchased from Solarbio Co. Ltd. (Beijing, China). Milli-Q Plus Ultra-pure water was used in protease assays (Millipore, Billerica, MA, USA). Reference compounds (purity > 98%), betaine, d-(-)-quinic acid, paeonol, chlorogenic acid, 3,4-dicaffeoylquinic acid, fraxetin, astilbin, isofraxidin, caffeic acid, rosmarinic acid, 7-hydroxycoumarine, alpinetin, and artemisinin, were purchased from Chendu De Keqi Rui Ke Biological Technology Co., Ltd. (Chendu, China). UPLC-QTOF-MS (Thermo Fisher Scientific, Bremen, Germany), Synergy II Microplate Reader (Biotec^®^ Instruments Inc. Winooski, VT, USA), and Analytical Balance MS205DU (Mettler-Toledo Co. riefensee, Zürich, Switzerland) were applied in this study.

### 3.3. Preparation of UPLC-MS Analysis Samples

The water extract (SGW), 30% MeOH extract (SG30), 60% MeOH extract (SG60), or 85% MeOH extract (SG85) were respectively reconstituted in water, 50% methanol, methanol, and isopropanol with a final concentration of 1.0 mg/mL for subsequent qualitative and quantitative analyses. All solutions were centrifuged (14,000× *g* for 10 min) before testing. Stock standard solutions (concentration: 1.0 mg/mL) of eight reference compounds (protocatechuic acid, chlorogenic acid, caffeic acid, fraxetin, isofraxetin, 3,4-dicaffeoylquinic acid, rosmarinic acid, and alpinetin) were separately prepared in methanol and stored at 4 °C. A mixed reference standard solution was prepared to get a series concentration of 100, 50, 25, 12.5, 6.25, 3.12, 1.56, 0.78, and 0.39 μg/mL. 

### 3.4. Fluorescence Resonance Energy Transfer (FRET) Assay 

Enzyme activities of HIV-1, Cathepsin L, and renin proteases were determined by a FRET assay, following the manufacturer’s protocol [17,18]. Briefly, the HIV-1 PR activity was assayed using 384-well microplates with a well volume of 120 μL (Corning Incorporated, Corning, NY, USA). Then, 8 μL of recombinant HIV-1 protease (wild type, NCBI accessiob: NC-001802, final concentration: 0.01 mg/mL (80 ng/well), AnaSpec Inc., Fremont, San Jose, CA, USA) was mixed with 2 μL of the test compounds (0.01~1.0 mg/mL). Reactions were then initiated by adding 10 μL of the HIV protease substrate (50X dilution with assay buffer). Fluorescence was measured using a EX/Em = 490/520 nm filter module on a Synergy II microplate reader. To evaluate the inhibitory activity of the test compounds towards Cathepsin L protease, fluorimetric SensoLyte^TM^ 520 Cathepsin L Protease Assay Kit (Lot#: AS72218-1010, AnaSpec Inc. San Jose, CA, USA) was used. Briefly, 2 μL test compound solutions (0.01~1.0 mg/mL) and 8 µL of human recombinant Cathepsin L protease (200× dilution with DTT-containing assay buffer, final concentration: 0.5 µg/mL) were added in the microplate wells. The microplate was pre-incubated at room temperature for 10 min. The enzymatic reaction was subsequently initiated by adding 10 μL of freshly diluted Cathepsin L substrate solution (100× dilution with assay buffer, final concentration: 0.01 mM) into each well. The microplate was gently shaken for 30 s before being incubated at 37 °C for 30 min. Fluorescence was measured at an emission of 520 nm and excitation of 490 nm. A Cathepsin L inhibitor, (100× dilution with assay buffer, final concentration: 1 µM) was used as a positive control. IC_50_ values of tested compounds were calculated by Microsoft Office Excel software using three different concentrations.

### 3.5. Data Acquisition with UPLC-MS

Major ingredients in active **SG** extracts were chemically characterized and quantified by UPLC-MS using a step gradient elution [0.1% formic acid/H_2_O(A)-0.1% formic acid/acetonitrile (B)] on a Phenomenex C_18_ column (2.1 × 100 mm, 1.7 μm, Phenomenex Kinetex, Torrance, CA, USA) equipped with an in-line filter. The gradient elution was as follows: 0–2 min, 2–20% B; 2–10 min, 20–31% B; 10–13 min, 31–100% B; 13–18 min, 100% B; 18–19 min, 100–2% B; and 19–22 min, 2% B. The sample injection volume was 2 μL, the flow rate was 0.3 mL/min, and the column temperature was 30 °C. The UPLC was coupled to the MS via a heated ESI source associated with a Q Exactive Plus, which was operating in full-scan and dd-MS^2^ for positive and negative polarities, respectively. For a full MS scan, the resolution was 35,000, the scan range was 100–1500 *m*/*z*, the ionization spray voltage was 4 kV, the sheath gas flow rate was 40 L/min, and the auxiliary gas flow rate was 10 L/min. For dd-MS^2^, the resolution was 17,500 with the isolation window of 3.0 *m*/*z* and normalized collision energy (NCE) of 10, 20, and 30. PRM mode was applied for the quantitative analysis of the eight components, with an injection volume of 10 μL and NCE of 20, 50, and 80.

### 3.6. Mass Spectrometry Data Analysis

Data analysis was performed using Compound Discoverer (Thermo Fisher Scientific, Shanghai, China) and relevant databases. The workflow includes retention time (RT) alignment, unknown compound detection, and compound grouping across all samples. Elemental compositions for all compounds were predicted and chemical backgrounds were hidden by using blank samples. Local database searches against mass lists (exact mass and RT) and mzVault spectral libraries were included in compound identification. A similarity search for all compounds with dd-MS^2^ data was performed using MassList and mzVault spectral libraries. MzLogic was applied to rank structures from MassList and mzVault spectral libraries search results. Xcalibur 4.2 was used for quantitative data processing.

### 3.7. Docking Studies

The molecular docking studies were carried out by using the ligand docking module of Schrödinger Suite (Schrödinger, LLC, New York City, NY, USA). The crystal structure of HIV-1 PR (PDB ID: 1QBS) and CatL PR (PDB ID: 3OF9) were chosen for docking studies. For each enzyme, the protein was prepared using the protein preparation wizard, which assigned bond orders, added hydrogen atoms, and removed water molecules. Subsequently, the protein was energy minimized by restrained minimization using a OPLS3 force field. The docking grid was generated at the orthosteric site, which was identified by picking out the original co-crystalized ligand using the receptor grid generation module. For the preparation of small molecule compounds, 3D structures were generated and energy-minimized by using the LigPrep module. The ligand docking module was used to dock the molecules to the orthosteric site of either HIV-1 PR or CatL PR, and Glide XP was chosen for docking precision [19].

## 4. Conclusions

COVID-19 patients usually experience a cough, a sore throat, fever, fatigue, a headache, diarrhea, septic shock, and coagulation disorder symptoms. Treatment of COVID-19 is relatively limited due to the mutation of SARS-CoV-2 and the shortage of specific antiviral drugs and vaccinations [20]. In clinic trials, single preparations of SGW have been used to treat acute respiratory infections, pneumonia, cellulitis, appendicitis shigellosis, thrombocytopenia, and malignancy diseases [8]. Our current results show that **SG** extracts (SGW, SG30) and their bioactive ingredients, chlorogenic acid, caffeic acid, rosmarinic acid, and alpinetin, possess moderate to strong inhibitory activities against CatL and/or HIV-1 PRs in vitro, implying that the **S****G** extracts could be used to prevent/treat SARS-CoV-2. The main dual inhibitors chlorogenic acid and rosmarinic acid could also be used as lead compounds in developing new antiviral agents. 

## Figures and Tables

**Figure 1 molecules-27-05552-f001:**
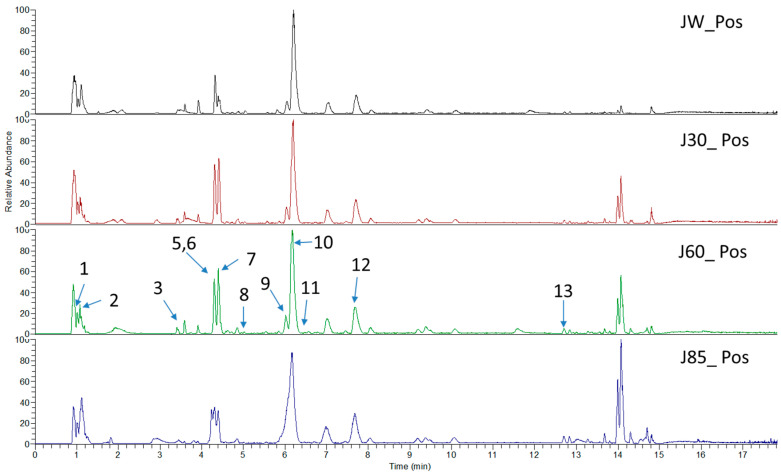
The chemical profile of **SG** extracts analyzed by UPLC-MS (positive). The numbers indicate the compounds.

**Figure 2 molecules-27-05552-f002:**
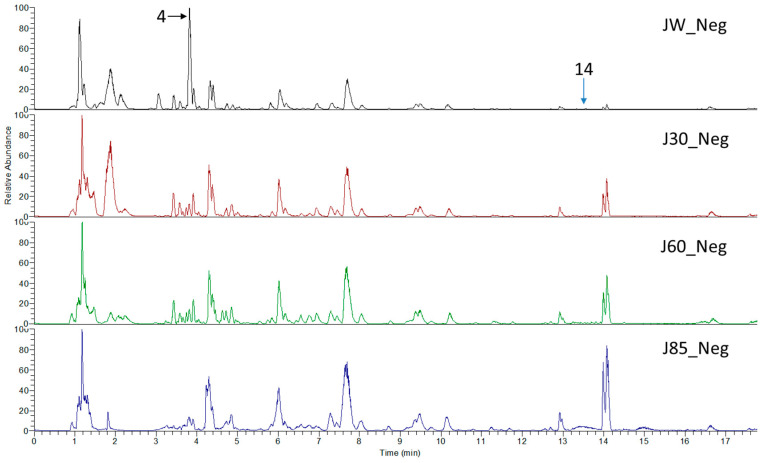
The chemical profile of **SG** extracts analyzed by UPLC-MS (negative). The numbers indicate the compounds.

**Figure 3 molecules-27-05552-f003:**
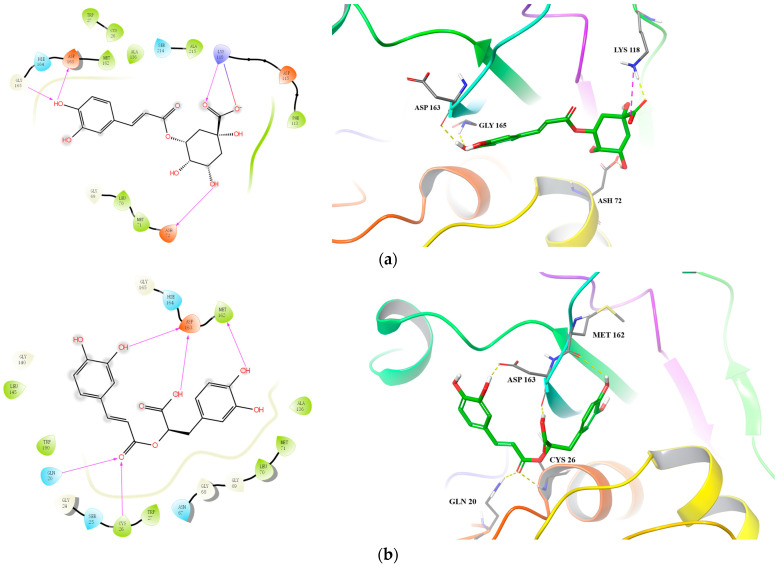
Predicted binding of chlorogenic acid (**a**) and rosmarinic acid (**b**) to CatL PR.

**Figure 4 molecules-27-05552-f004:**
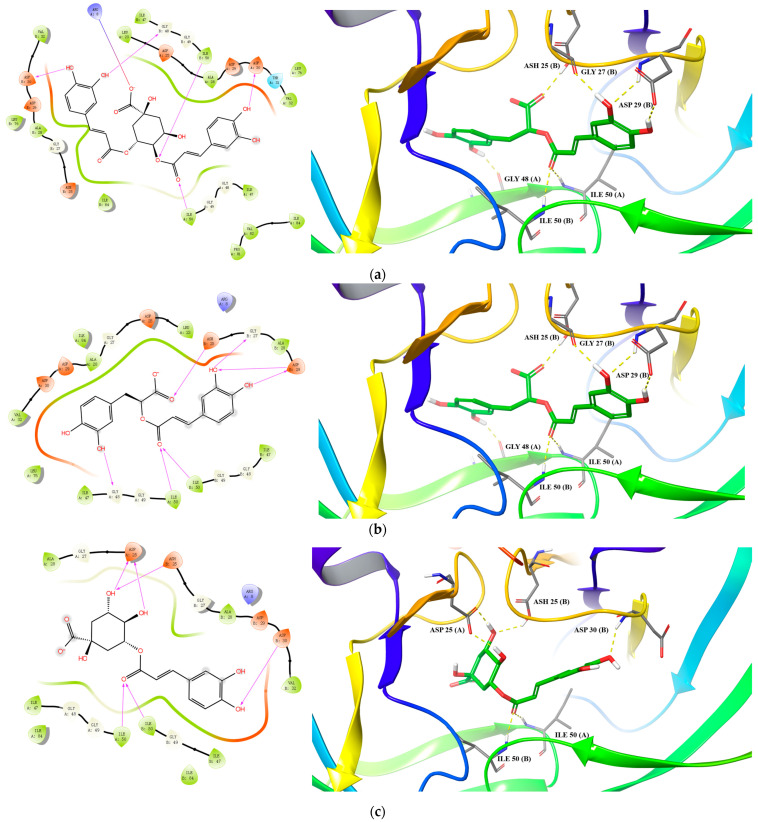
Predicted binding of 3,4-dicaffeoylquinic acid (**a**), rosmarinic acid (**b**), and chlorogenic acid (**c**) to HIV-1 PR.

**Table 1 molecules-27-05552-t001:** The IC_50_ values of *SG* extracts against HIV-1, Cat L, and renin PRs (*n* = 3).

Name	IC_50_ ± RSD% (mg/mL)		
	HIV-1 PR	CatL PR	Renin PR
SGW	0.003 ± 0.058%	0.26 ± 2.38%	0.13 ± 1.2%
SG30	0.008 ± 0.034%	0.24 ± 0.87%	0.12 ± 1.6%
SG60	0.063 ± 0.71%	0.15 ± 1.06%	0.13 ± 2.8%
SG85	0.07 ± 0.74%	0.21 ± 1.19%	>100
PC1	0.25 ± 0.03% (μM)	-	-
PC2	-	0.0032 ± 0.044% (μM)	-
PC3	-	-	0.95 ± 0.021% (μM)

-: no test. PC1: pepstatin A, positive control for HIV-1 protease. PC2: Cathepsin L inhibitor, positive control for Cathepsin L protease. PC3: positive control for renin protease.

**Table 2 molecules-27-05552-t002:** The main active ingredients of **SG** extracts.

No.	R_t_ [min]	Name	Structures	Formula
**1**	1.12	Betaine	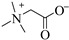	C_5_H_11_NO_2_
**2**	1.19	D-(-)-Quinic acid	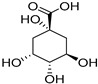	C_7_H_12_O_6_
**3**	3.47	Paeonol	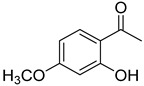	C_9_H_10_O_3_
**4**	3.81	Protocatechuic acid	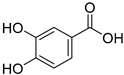	C_7_H_6_O_4_
**5**	4.30	Chlorogenic acid	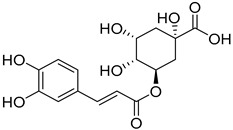	C_16_H_18_O_9_
**6**	4.32	7-Hydroxycoumarine	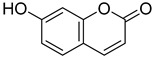	C_9_H_6_O_3_
**7**	4.74	Caffeic acid	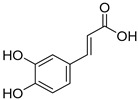	C_9_H_8_O_4_
**8**	5.02	Fraxetin	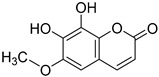	C_10_H_8_O_5_
**9**	5.96	Astilbin	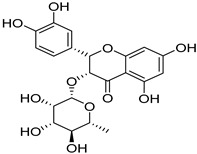	C_21_H_22_O_11_
**10**	6.04	Isofraxidin	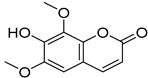	C_11_H_10_O_5_
**11**	6.59	3,4-Dicaffeoylquinic acid	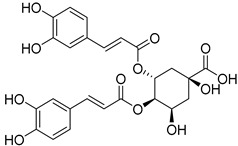	C_25_H_24_O_12_
**12**	7.71	Rosmarinic acid	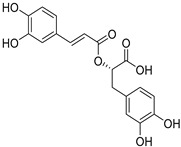	C_18_H_16_O_8_
**13**	12.65	Alpinetin	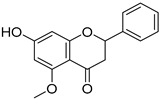	C_16_H_14_O_4_
**14**	13.64	Artemisinin	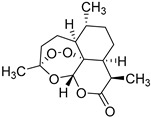	C_15_H_22_O_5_

**Table 3 molecules-27-05552-t003:** Inhibition of CatL PR by major ingredients of **SG** extracts (*n* = 3).

No.	Name	CatL PR	
		Inhibition (%) ± SD at 0.01 (mg/mL)	IC_50_ ± RSD% (mg/mL)
**4**	Protocatechuic acid	42.1 ± 2.25	>100
**5**	Chlorogenic acid	40.8 ± 2.54	>100
**7**	Caffeic acid	32.1 ± 1.68	>100
**10**	Isofraxidin	27.0 ± 2.37	>100
**6**	7-Hydroxycoumarine	22.6 ± 0.53	>100
**12**	Rosmarinic acid	19.2 ± 1.70	>100
**8**	Fraxetin	15.0 ± 5.25	>100
**13**	Alpinetin	11.9 ± 8.99	>100
**2**	D-(-)-Quinic acid	−0.32 ± 8.61	>100
**14**	Artemisinin	−1.1 ± 0.52	>100
**3**	Paeonol	−15.0 ± 0.56	>100
**1**	Betaine	−15.5 ± 5.76	>100
**9**	Astilbin	−24.7 ± 8.14	>100
**11**	3,4-Dicaffeoylquinic acid	−31.3 ± 6.24	>100
PC		99.2 ± 0.13 (0.02 µM)	0.0032 ± 0.044% (μM)

**Table 4 molecules-27-05552-t004:** Inhibition of HIV-1 PR by major ingredients of **SG** extracts (*n* = 3).

No.	Name	HIV-1 PR	
		Inhibition (%) ± SD at 1.0 (mg/mL)	IC_50_ ± RSD% (mg/mL)
**5**	Chlorogenic acid	100.0 ± 3.75	0.026 ± 1.03%
**11**	3,4-Dicaffeoylquinic acid	86.1 ± 3.92	0.13 ± 2.84%
**7**	Caffeic acid	65.9 ± 0.00	0.48 ± 0.124%
**12**	Rosmarinic acid	68.8 ± 4.21	0.54 ± 0.55%
**9**	Astilbin	68.4 ± 5.45	0.54 ± 4.34%
**13**	Alpinetin	62.3 ± 0.96	0.62 ± 3.44%
**14**	Artemisinin	50.8 ± 0.58	0.95 ± 2.77%
**4**	Protocatechuic acid	47.9 ± 1.82	>100
**6**	7-Hydroxycoumarine	28.9 ± 4.41	>100
**8**	Fraxetin	24.1 ± 6.70	>100
**1**	Betaine	8.95 ± 4.61	>100
**10**	Isofraxidin	−2.09 ± 8.57	>100
**3**	Paeonol	−12.6 ± 0.26	>100
**2**	D-(-)-Quinic acid	−20.1 ± 2.61	>100
Pepstatin A			0.25 ± 0.03% (µM)

**Table 5 molecules-27-05552-t005:** Docking main bioactive compounds from *Sarcandra glabra* extracts against CatL PR. (PDB ID: 3OF9).

No.	Name	Docking Score	Glide Gscore	Glide Emodel (kcal/mol)
**5**	Chlorogenic acid	−8.282	−8.285	−48.119
**12**	Rosmarinic acid	−7.616	−7.616	−58.310
**4**	Protocatechuic acid	−6.043	−6.043	−26.518
**8**	Fraxetin	−5.335	−5.356	−30.303
**7**	Caffeic acid	−4.766	−4.766	−30.038
**6**	7-Hydroxycoumarine	−4.635	−4.645	−27.908
**13**	Alpinetin	−3.913	−3.913	−36.021
**10**	Isofraxidin	−3.258	−3.316	−28.556
PC	Cathepsin L inhibitor	−7.822	−7.823	−93.170

**Table 6 molecules-27-05552-t006:** Docking active compounds from **SG** extracts against HIV-1 PR (PDB ID: 1QBS).

NO.	Name	Docking Score	Glide Gscore	Glide Emodel (kcal/mol)
**11**	3,4-Dicaffeoylquinic acid	−14.079	−14.079	−99.795
**12**	Rosmarinic acid	−11.268	−11.268	−72.868
**5**	Chlorogenic acid	−10.679	−10.682	−75.647
**9**	Astilbin	−9.267	−9.283	−77.052
**7**	Caffeic acid	−7.071	−7.071	−35.043
**13**	Alpinetin	−6.765	−6.765	−46.293
**14**	Artemisinin	−4.971	−4.971	−37.755
PC	Pepstatin A	−10.940	−10.941	−131.591

**Table 7 molecules-27-05552-t007:** The contents of main active ingredients in different **SG** extracts.

No	Rt (min)	Name	SGW	SG30	SG60	SG85	Regression Equation
			Content (μg/mg)				
**4**	3.81	Protocatechuic acid	14.94	0.40	0.39	0.51	Y = −3.92 × 10^−3^X^2^ + 4.938 × 10^2^X − 1.197 × 10^5^; R^2^ = 0.9998
**5**	4.30	Chlorogenic acid	12.724	15.668	15.094	15.767	Y = −5.403 × 10^−2^X^2^ + 1.143 × 10^4^X + 1.748 × 10^4^; R^2^ = 0.9999
**7**	4.74	Caffeic acid	1.712	1.823	2.296	2.468	Y = −3.553 × 10^−1^X^2^ + 1.433 × 10^4^X + 1.123 × 10^6^; R^2^ = 0.9996
**8**	5.02	Fraxetin	0.020	0.004	0.011	0.022	Y = −2.997 × 10^−1^X^2^ + 4.775 × 10^4^X + 1.02 × 10^7^; R^2^ = 0.9999
**10**	6.04	Isofraxetin	4.800	4.260	3.925	4.995	Y = −9.136 × 10^−2^X^2^ + 1.43 × 10^4^X + 3.492 × 10^6^; R^2^ = 0.9993
**11**	6.59	3,4-Dicaffeoylquinic acid	5.215	6.845	6.848	7.813	Y = −3.886 × 10^−2^X^2^ + 1.057 × 10^4^X + 4.053 × 10^6^; R^2^ = 0.9996
**12**	7.71	Rosmarinic acid	21.242	22.804	27.730	29.943	Y = −3.942 × 10^−3^X^2^ + 1.423 × 10^3^X − 8.076 × 10^5^; R^2^ = 1.0000
**13**	12.65	Alpinetin	0.026	0.051	0.059	0.113	Y = −1.703 × 10^−1^X^2^ + 4.281 × 10^5^X + 8.059 × 10^7^; R^2^ = 0.9998

## Data Availability

The data presented in this study are available in this article and on request from the author and corresponding author.

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
