# Peer review of "Dual Inhibition of HIV-1 and Cathepsin L Proteases by *Sarcandra glabra"

_molecules, 2022, doi:10.3390/molecules27175552_

Round 1
Reviewer 1 Report
It is a solid report on of anti-viral activity of plant extracts. The materials and methods are well presented; however, the results are not. It is not clear how the measured data (inhibition, IC50) are connected to docking results. A suggestion is to present the chemical compounds in hierarchical order from the most active to less active for all measurements and for docking. Also, the connection from individual compounds to SG extracts (SGW, SG30, etc.) is poorly presented.
Reviewer 2 Report
Although the concept and study method with results is qualified, there are some points that need to be further discussed:
Point 1:
Regarding interaction between potential substances and Cathepsin L protease according to in vitro, is it reversible or irreversible? Why the authors did not evaluate the reversible inhibition of Cathepsin L protease? If a substance is an irreversible inhibitor, are there any risk of toxicity? The author must further discuss.
Point 2:
In docking step, how can we define a good binding complex by docking score? Why did the study select -6.0 kcal/mol as a threshold in binding toward Cathepsin L protease while this figure for HIC-1 PR was -9.0 kcal/mol (different value)?
Point 4:
How did the author analyze a salbridge between ligand and protein. It must be explain in detail.
Point 5:
The authors must give more information about the docking process on the method.
Point 6:
There are many mistakes in grammar and typos. The author must carefully checked before re-submit the manuscript.
Round 2
Reviewer 1 Report
Authors adequately answered the questions.
Reviewer 2 Report
Point 3: the author need to define "salt bridge" interaction term before give the examples.